# A *Review* of Effects of Environment on Brain Size in Insects

**DOI:** 10.3390/insects12050461

**Published:** 2021-05-17

**Authors:** Thomas Carle

**Affiliations:** Faculty of Biology, Kyushu University, Fukuoka 819-0395, Japan; th.carle@gmail.com

**Keywords:** brain size, environment, evolution

## Abstract

**Simple Summary:**

What makes a big brain is fascinating since it is considered as a measure of intelligence. Above all, brain size is associated with body size. If species that have evolved with complex social behaviours possess relatively bigger brains than those deprived of such behaviours, this does not constitute the only factor affecting brain size. Other factors such as individual experience or surrounding environment also play roles in the size of the brain. In this review, I summarize the recent findings about the effects of environment on brain size in insects. I also discuss evidence about how the environment has an impact on sensory systems and influences brain size.

**Abstract:**

Brain size fascinates society as well as researchers since it is a measure often associated with intelligence and was used to define species with high “intellectual capabilities”. In general, brain size is correlated with body size. However, there are disparities in terms of relative brain size between species that may be explained by several factors such as the complexity of social behaviour, the ‘social brain hypothesis’, or learning and memory capabilities. These disparities are used to classify species according to an ‘encephalization quotient’. However, environment also has an important role on the development and evolution of brain size. In this review, I summarise the recent studies looking at the effects of environment on brain size in insects, and introduce the idea that the role of environment might be mediated through the relationship between olfaction and vision. I also discussed this idea with studies that contradict this way of thinking.

## 1. Introduction

Brain is in perpetual change, through evolution between and within species or even through the life of individuals. These changes do not only comprise specific neuronal structures that play specific functions (e.g., visual cortex, thalamus…) and are specific to species, but also changes in neuronal networks and mechanisms. As a better comprehension of brain is fundamental in various domains such as evolutionary biology, psychology or ethology, to study brain has fascinated many researchers. In particular, researchers have been fascinated by its size since it could have been used to compare the level of intelligence between species or individuals. In this review, I will brush off this idea by resuming studies investigating effects of environment on brain size in insects.

## 2. Brain Size

What makes a brain big? The idea that a bigger brain is directly related to better intelligence is biased on the fact that people naively and simply associate a bigger brain with a larger number of neurons and, therefore, a greater number of neural connexions. However, many factors nullify this simplistic point of view, such as the size of neurons [1]. Before talking about intelligence, Jerison [2] showed a general allometric relationship between brain size and body size across mammals. In other words, it means that brain size is primarily dependant to body size, and was first found to follow the equation: log (Brain weight) = 0.76 × log (Body weight) −1.28 [3,4], or more recently (50 years later and using substantially more species) log (Brain weight) = 0.75 × log (Body weight) −1.26 [5]. Looking more attentively at Jerison’s study [2], there are disparities in this linear relationship between the logarithmic values of brain and body sizes between and within species: some species have a relative brain size larger than other species. In addition, when analysing this relationship between brain and body size by animal orders, the slope varies ranging from 0.24 to 0.81 [5].

These deviations have sometimes been used to estimate an animal’s cognitive abilities in comparison to other species. For example, the presence or absence of complex cognitive capabilities, which might be required in social behaviours, can affect brain size as posited by the ‘social brain hypothesis’ [6,7,8,9,10,11]. This idea persisted until recent studies suggested that ecological drivers, including diets, are better predictors of brain size than sociality in primates [12,13] and insects [14]. More generally, apart from ecological factors [15], other hypotheses considering energetic [16], life history [17] and behavioural factors [18] have been highlighted to explain these disparities and to support anticipated criticisms about the relation between brain size and intelligence [19,20,21,22].

The purpose of this review is not to question brain size due to the evolution of cognitive capabilities such as those involved in social behaviour or learning and memory capabilities, but to highlight the idea that environment also plays a role in evolution and development of brain size through a summary of studies done in insects and a parallel with mammals.

## 3. Haller’s Rule, Brain Size in Insects

In 1762, A. von Haller predicted that smaller animals have relatively larger brains than larger-bodied forms, a concept known as ‘Haller’s rule’ [23]. This prediction is easily verifiable by the value of the slope that Armstrong [3,4] and Burger et al. [5] found in their equation. The slope being inferior to 1 means that the increase in brain weight is inferior to the increase in body weight during brain evolution and, as a result, smaller species possess relatively bigger brains. To take a concrete example, the brain takes a large portion of the body mass in insects, about 16% [24,25], whereas it takes about 2–2.5% of body mass in humans [26,27] and about 0.07–0.18% in elephants [28,29]. However, the brain–body mass ratio in adult mice is close to 1–1.6% [30,31], which is inferior to that seen in humans and seems to contradict Haller’s rule at first sight. This may be explained if humans are considered to be relatively more intelligent than mice. 

To take this argument further, we need to consider if ‘intelligence’ can be anatomically measured. To find a comparative anatomical measure of “intelligence” between species, Jerison and Barlow [32] proposed the ‘encephalization quotient’ (EQ): the ratio between brain mass observed and that predicted from body size within an animal of a given species. The EQ has been evaluated to equal 7.4–7.8 in humans, 1.3 in African elephants and 0.5 in mice [28]. This means that the human brain is 7–8 times larger than expected, whereas in mice, it is 0.5 times larger (or twice smaller) than expected. This may explain why elephants have a comparatively small brain–body ratio despite being considered as quite intelligent animals. It also shows that the human brain is disproportionally bigger compared to other species and explains why its relative size is superior to that in mice. However, the link between EQ and “intelligence” is still in debate [28], and recent ways of thinking tend to consider the number of neurons rather than the mass as a better indicator of cognitive capabilities see [33].

In general, insect brain size follows Haller’s rule (Figure 1). This relationship is verifiable both between species [25,26,34], and within species [24,26,34,35]. However, recent studies have pointed to several exceptions to Haller’s rule [36,37]. For example, Van der Woude et al. [37] performed an intraspecific comparison between differently sized individuals of the small parasitic wasp *Trichogramma evanescens*, and showed for the first time an isometric brain–body size relation, so brain size is directly proportional to body size (with the brain occupying 8.2% of the body in this case). The authors explained their observations by the fact that an increase in expensive brain tissue in the smallest individuals would be too costly in terms of energy expenditure and, therefore, there would be a selective pressure for maintaining smaller brains in smaller individuals. In the line of this study, Groothius et al. [36] made the same observation in the wasp *Nasonia vitripennis*. However, in this case, the authors observed an isometric relationship for the smallest wasps, but not on the largest individuals. To date, these two studies provide the only exceptions to Haller’s rule, which might be associated with these wasps’ parasitic lifestyle. 

## 4. Effect of Environment on Brain Size in Vertebrates

As mentioned previously, whilst body size and factors such as increase in cognitive [7,38,39] or energetic demands [40,41,42,43] have driven the evolution of brain size in vertebrates, environment also plays an important role [12,44]. By environment, I refer to external factors that impact the development and evolution of brain size such as landscapes (including the presence or absence of light), climate, presence or absence of conspecifics or predators, or scarcitity of food. In particular, how environment modulates investment in sensory rather than cognitive systems is an important determinant of brain size [45,46]. 

One of the most extreme environments that we can think about that has an important impact on brain size is darkness. It is well known that whilst crepuscular and nocturnal species tend to have evolved very specialised eyes for making the most of low light levels [47,48], those living in aphotic environments have been selected to have much smaller or non-functional eyes [49,50,51,52]. Perhaps not surprisingly, given that maintaining neural circuits is energetically expensive [40,43], it is not only the size of eyes that reduced, but also areas of brain in association with processing visual information [45,49,50,52]. Furthermore, in association to a reduced visual system [40,53], cave or other dark-dwelling species generally have reduced relative brain size [42,54]. 

Changes induced by darkness do not only occur on an evolutionary scale, but may be seen within development of individual animals. A long history of experiments on individuals reared in the dark, or with their eyelids sutured, has associated the lack of visual stimulation with deficiencies in neural mechanisms at the retina level [55,56] and in brain areas dedicated to vision, such as the primary visual cortex [57,58,59,60] and lateral geniculate body [61,62,63]. In particular, there is a reduction in the number of retinal cells [61] and in the number of neuronal connexions [59,62]. 

Although darkness constitutes an environment where selective pressure is important, a large number of studies showed that enriched [64,65,66,67], changed [15,68], predatory ([69,70] but [71,72]), rearing [73], and social environment [74] all impact brain size for review [75]. For example, in a very recent study, Fong et al. [76] found that guppies (*Poecilia reticulata*) reared in an enriched environment (a spatial-learning environment where fish daily experienced a maze to find food) possess a larger brain associated with a larger optic tectum compared with fish that did not experience the presence of a maze in their tank. All these studies support Hebb’s hypothesis [77], which predicts that training, or differential experience, induces neurochemical changes in the cerebral cortex, and a differential cortex weight between those individuals that received this training, or experience, and those that did not [78].

## 5. Effect of Environment on Brain Size in Insects

Environment also plays an important role in brain evolution and development in insects. Technau [79] was the first to investigate the effects of environment on insects’ brain size and showed that individual experience impacts brain size (Table 1). Technau counted the number of fibres in the mushroom bodies (higher order processing areas in the central brain that integrates information from the visual and olfactory systems) of fruit flies (*Drosophila melanogaster)* and found that the number of fibres reduced when individual flies experienced social isolation or were deprived of their antennal input. Similarly, Heisenberg et al. [80] found that that larval density (which might affect social experience) had a positive effect on the number of Kenyon cell fibres (intrinsic neurons of the mushroom bodies), the absolute volume of mushroom body calyces and of optic lobe medulla and lobula (brain areas processing visual information) when these flies become adult. However, these results contrast with those obtained recently by Wang et al. [81] (Figure 2), who found that extreme larval crowding reduced the volume of brain areas such as the optic and antennal lobes (brain areas processing visual and olfactory information, respectively), and central complex (neuropils in the centre of the insect brain that play a crucial role in spatial aspects of sensory integration and motor control). An explanation may be found in the density of flies used by both studies: whereas Heisenberg et al. [80] compared densities between 5–6 to 20–25 flies/cm^2^, Wang et al. [81] tested densities going to 60 flies/cm^2^. In this case, it could be interesting to ensure that competition for food access does not affect brain development since a lack of food might affect body and brain sizes. 

In another type of population density, Ott and Rogers [82] showed that gregarious desert locusts have substantially larger brains compared with locusts in the solitarious phase (Figure 2). However, whereas larger sizes were observed in all sub-compartments (optic lobes, antennal lobes, midbrain, mushroom bodies, etc.), the relative size of optic lobes and antennal lobes reduced in size whereas the mushroom bodies (calyx and lobes) and central complex increased in size. In this specific case, the authors concluded that the larger brains of gregarious locusts prioritize higher integration, which may support the behavioural demands of generalist foraging and living in dense and highly mobile swarms dominated by intense intraspecific competition. This shows that, even in insects, brain plasticity does not occur only during the development of individuals, but may happen during lifetime, as very recently shown [99].

In addition to very high densities and heat stress, a single daily 39.5 °C pulse for 35 min had a negative impact on brain size in fruit flies (*D. melanogaster*). The volumes of the antennal lobes [83] and mushroom bodies (particularly of the calyx and pedunculus) [81] were reduced, but the optic lobes or central complex were not affected [81,83] (Table 1, Figure 2).

There is a combination of age and experience acquired during life playing a role on brain size. Fahrbach et al. [100] found that all regions of the mushroom bodies increase in size (with the exception of the basal ring) during the first week of life in honeybees (*Apis mellifera*) that were kept in total darkness and in social isolation, in other words, without experiencing visual and social environments. Although Fahrbach et al. [100] found that age affect brain size, previous experiments from Withers et al. [84] and Durst et al. [85] had already concluded that this difference of size did not depend only on the age of the bees but on their experience. In particular, Withers et al. [84] analysed the anatomical changes in association with age and the naturally occurring behavioural transition from nursing to foraging tasks, and showed that the size of brain structures depended on foraging experience in honeybees (Table 1, Figure 2). They found that nurses possess relatively bigger olfactory glomeruli, spherical neuropils whose number is more and less related to the number of different types of olfactory receptors, than foragers despite the fact that foragers also depend heavily on olfaction. While Withers et al. [84] did not find a significant increase in the size of the protocerebrum (including mushroom bodies, accessory protocerebrum and neuronal somata) between the nurses and foragers, Durst et al. [85] found that foragers possess bigger mushroom bodies compared with nurses in age-controlled bees. The difference between these studies might be due to the fact that Durst et al. [85] measured the volume of the different mushroom body sub-compartments whereas Withers et al. [84] measured the whole mushroom body neuropil region. Another explanation might lie to the age of the bees used: 11 days old nurses and 11 days old foragers in Durst et al.’s [85] study and Withers et al. [84] determined that foragers were almost certainly 7–10 days older than nurses. However, age is not always related to an increase of brain size. Julian and Gronenberg [86] found in ants (*Messor pergandei*) that the brains of mature queens are significantly smaller than those of virgin females at the time of their mating flight. This reduction seems associated to reduced behavioural repertoires and a life in the dark. 

In general, light or dark conditions have an important impact on the development and evolution of brain size. For example, Barth showed that rearing fruit flies (*D. melanogaster*) in the dark decreases not only the size of the adult visual system [87], but also of mushroom bodies and central complex [88]. These decreases seems associated with changes in visually guided choice behaviour compared to controls reared in a normal light/dark cycle [101] (Table 1). Inversely, Stieb et al. [89] observed that light exposure, which occurs during behavioural transition from brood care and food processing in the nest to outdoor foraging in ants (genus *Cataglyphis*), triggers mushroom bodies’ calycal growth associated with a reduction in microglumeruli numbers in the visual and olfactory input regions of the calyx. These studies seem to contrast with Jones et al.’s [90] study on worker bumble bees (*Bombus impatiens*) that showed that rearing individuals in darkness causes mushroom body size to increase. 

In terms of evolutive processes, Özer and Carle [91] recently tested how visual enrichment (normal 12:12 L:D lighting conditions) affects brain size by rearing Dark-flies (a strain of *D.*
*melanogaster* reared in the dark since 1954 [102]) (Table 1, Figure 2). They observed that sizes of both whole brain and optic lobes increased significantly in size after 65 generations, whereas that of antennal lobes decreased [91,103]. This seems in line with Montgomery et al.’s [94] (Figure 2) and Sheehan et al.’s [95] observations showing that nocturnal species (moths and ants, respectively) invest relatively less in the primary visual processing regions, but relatively more in both the primary olfactory processing regions. However, these two studies discuss evolutionary differences between species [94,95], whereas the other experience related plasticity within the same species [91]. In Sheehan et al.’s [95] observations, they also found that nocturnal ants invested relatively more in the integration centres of visual and olfactory sensory information and possess bigger mushroom bodies, which is in accordance with Jones et al.’s [90] study in bumblebees. Özer and Carle [91] and Jones et al. [90], all agree on the fact that visual enrichment in species that lived in the dark has a negative effect on the size of the olfactory system. To be more precise, the antennal lobes are bigger when individuals stay in the dark [90,91]. Decreased size of antennal lobes in the presence of visual stimuli in individuals habituated to darkness might be linked to the parallel development of visual system. Although the mechanism has stayed unknown until now, it recently guides researchers to the notion of trade-off between vision and olfaction.

## 6. Balance Between Vision and Olfaction

The idea that there would be a trade-off between vision and olfaction was introduced in 1995 with observations of brain in primates [45]. A trade-off between vision and olfaction means that the size of visual and olfactory brain structures are inversely correlated. In their pioneering study, Barton et al. [45] observed this negative correlation in primates, but not in bats or other mammalian species that are insectivores. After pursuing his investigations on primates, Barton [104] showed three years later that in primates, the bigger the visual system, the bigger the brain and, therefore that brain size is associated with visual specialization. However, he concluded on the fact that nocturnal frugivores may rely more on olfaction, but that the larger size of olfactory structure (and possibly auditory structures) in nocturnal species “seems” to have offset the smaller size of their visual structures in primates.

Since Barton’s observations, the idea that there is a trade-off between vision and olfaction has recently been explored in insects [91,94,95,96,103,105]. For example, Montgomery et al. [94] showed that the levels of sensory investment in the diurnal glasswing butterfly (*Godyris zavaleta*) is intermediate between the diurnal monarch butterfly (*Danaus plexippus*), which invests heavily in visual neuropil, and night-flying moths, which invest more in olfactory neuropil. In other words, the glasswing butterfly has relatively larger antennal lobes and smaller optic lobes compared with the monarch butterfly. These observations at the neural level are consistent with behavioural observations, suggesting that odours may be more important in guiding a suite of behaviours in the glasswing than in the monarch [94]. Indeed, compared to monarch butterfly, *Godyris zavaleta* has a derived mating behaviour where the males use pyrrolizidine alkaloids both as precursors for pheromone synthesis and for chemical protection. The link between neural size and behavioural preference for a sensory system was demonstrated one year later by Stöckl et al. [96]. By comparing two closely related hawk moth species, *Macroglossum stellatarum* and *Deilephila elpenor*, the authors found that the use of visual or olfactory cues in a foraging task is correlated with the neural investment for the visual or olfactory system in these species. This differential investment observed between nocturnal and diurnal species was also observed in congeneric species of the Australian bull ant (*Myrmecia*). In their study, Sheehan et al. [95] showed that the nocturnal ants invested relatively less in the optic lobes, but relatively more in the antennal lobes compared to diurnal species. Beyond comparing two or three different species, which may constitute specific cases, Keesey et al. [105] compared a large number of fruit flies species (62 species) and showed recently that the size of antennal lobes and optic lobes are inversely proportional to each other. This constitutes a main progress, since this inversed relationship seems not exclusively dependent on diurnal and nocturnal selective pressures.

## 7. Effect of Environment on Brain Size through This Vision and Olfaction Trade-Off

Darkness does not only play a role on investment between the visual and olfactory systems [94,95,96], but seems also to impact brain size. Within the same species, Özer and Carle [91] observed that Dark-flies reared in normal lighting conditions evolved with larger optic lobes and smaller antennal lobes, but also with larger whole brain. Inversely, Dark-flies possess smaller whole brain compared to the Oregon-R-S strain at the origin of the Dark-flies [91,103]. In addition, flies reared in the dark have a smaller volume of optic lobes, mushroom bodies and central complex compared to flies reared in normal lighting conditions (12:12 L:D cycle) [87,88]. From these observations, it seems that there is a link between the size of the visual system and brain size. Such correlation has been observed not only in invertebrates [91,98,106], but also in vertebrates [46,104,107]. Therefore, we may think that increase or decrease of brain size might be directly associated to the number of visual inputs (neuronal efferents coming from the optic lobes). 

In such a perspective, we can easily imagine that, not only dark environment, but also other environments affecting this trade-off play a role on brain size. In particular, environments that favour the use of olfaction to the detriment of vision, such as dense forests, might be at the origin of a reduction of brain size. For example, Stieb et al. [97] showed that wood ant species form the Formica species (*F. rufibarbis* and *F. sanguinea*) possess a higher number of olfactory glomeruli compared with desert ant species from the Cataglyphis species (*C. fortis*, *C. albicans*, *C. bicolor*, *C. rubra*, and *C. noda*). This is likely to reflect the importance of olfaction in wood ant species and in species living in woods in general. In woods, due to the presence of trees and leaves, to get visual information at a distance is obstructed by the presence of these obstacles. Therefore, relying on smell seems an evolutionarily better strategy than developing a more sophisticated visual system for detecting sexual partners, potential food or simply for navigating. Conversely, desert ant species are famous for the study of animal navigation and visually guided behaviours [108]. In this case, we may expect that desert ant species possess a bigger brain than wood ant species. This is hypothetically confirmed with the fact that ants from the Cataglyphis species (*C. bicolor*, *C. maurtanica*, and *C. viatica)* seem to possess bigger brains compared to those from the Formica species (*F. japonica*) [37]. However, the species used in these studies are not exactly the same, and this conclusion remains speculative. Therefore, there is a new avenue of research in the future by investigating effects of dense environments on evolution of brain size. 

## 8. Vision and Olfaction Trade-Off A Universal Rule

However, to say that this trade-off is at the base of evolution of brain size would be too conclusive. For instance, In another example, although the authors themselves discussed this conclusion, Muscedere et al. [109] directly contested the existence of a trade-off between vision and olfaction after measuring brain size and the size of sensory systems in diverse eusocial hymenopteran species (ants, bees and wasps). In another example, Sheehan et al. [95] effectively found a negative correlation between optic lobes and antennal lobes comparing congeneric diurnal and nocturnal ants, but the relative brain size did not differ between them. One reason might come from the fact that the ratio between optic and antennal lobe volumes in ants is very low. This means that brain size is less dependent on efferent coming from the visual system compared to insect species where the relative size of optic lobes (relative to central brain) is important. Therefore, investing in vision to the detriment of olfaction, or inversely, might have no effect on brain size in ants. In this case, brain size between desert and wood ant species might be similar, invalidating the hypothesis that dense forests have a negative impact on brain size. However, if this hypothesis is not verified in ants, it needs to be validated or invalidated for other species having a larger investment in the visual system. Another reason, which may explain the lack of difference in brain size between diurnal and nocturnal ants, might be the presence of other unidentified factor(s) that have counterbalanced the negative effect of darkness on brain size. 

Furthermore, some species are anosmic. In other words, these species do not possess the sense of smell, as with cicadas. This constitutes a specific case where the trade-off between vision and olfaction has no sense. If it is well known that cicadas are deprived of smell and antennal lobes. They are also deprived of the presence of calyx in the mushroom bodies [110,111]. We may therefore think that whole brain size is also reduced. However, to my knowledge, cicadas’ brain size if far from being known. If cicadas may compensate for the lack of olfaction with sound, how this adaptation shapes and affects brain size is far from being known. 

Refuting the trade-off between vision and olfaction does not work only in cicadas. As previously mentioned, Barton et al. [45] did not find a negative correlation between vision and olfaction in bats and insectivores. In this case, investment in other sensory modalities might have an impact on this bilateral negative correlation. For example, it is well known that bats mainly rely on echolocation [112], and other species mainly rely on somatosensory information such as the star-nose mole [49] or naked mole rats [113]. In these cases, it would be interesting to know if investing in one specific sensory modality is to the detriment to another ones and how it affects brain size in general. However, if measuring the volume of optic and antennal lobes enables the determination of the sensory investment in insect species for vision and olfaction, respectively, how to measure investment in other sensory modalities (audition or mechanoreception) remains a challenge and needs to be discover in the future for a better comprehension of multisensory ecology. 

## 9. Conclusions

In this review, I highlighted recent advances showing that environment, certainly through the balance between vision and olfaction, plays a role on relative brain size, as seen in fruit flies and primates [91,104]. However, this does not mean that brain size is mostly dominated by the relationship between vision and olfaction. If visual specialization may have a positive impact on brain size in closely related species, it is only one factor among others when comparing all species. Emergence of better learning and memory capabilities, as well as more complex social interactions are also factors that drove evolution of brain size. One challenge in the future will be to take into account all these factors at the same time in comparative studies of brain size between species. To this goal, insect studies may provide good advances in this domain because of their relatively simple nervous system and the presence of specific cases, such as cicadas. However, with such a perspective, it is now necessary to take into account other sensory systems to determine their impact on brain size, and the initial challenge here will lie in finding clear measures reflecting sensory investment in modalities such as audition and mechanoreception in insects. 

I presented many studies showing that factors rather than intelligence are associated to variations in brain size and confirm recent criticisms. The EQ might be a better indicator of a combination between investment in the visual system, social behaviour or learning and memory capabilities in species rather than being simply a comparative anatomical measure of “intelligence”. Furthermore, none must forget that brain and the sensory systems themselves have evolved sharing common design principles during evolution [114,115]. It is the neuron itself that evolved [116], in particular in terms of size [1]. Therefore, larger brains do not automatically mean a larger number of neurons if neuronal size increases within evolution and, as a consequence, a larger brain is not automatically associated with a higher level of neural connexion. Famous groups of researchers lead by Giurfa and Menzel have brilliantly showed that, despite their size, mini-brains exhibit unsuspected complex cognitive capabilities [117,118,119].

## Figures and Tables

**Figure 1 insects-12-00461-f001:**
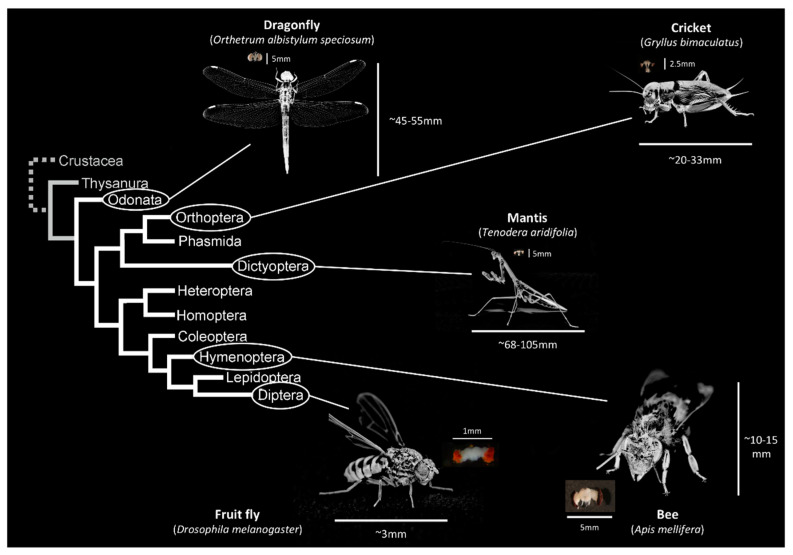
Illustrations of brains across insects’ phylogeny.

**Figure 2 insects-12-00461-f002:**
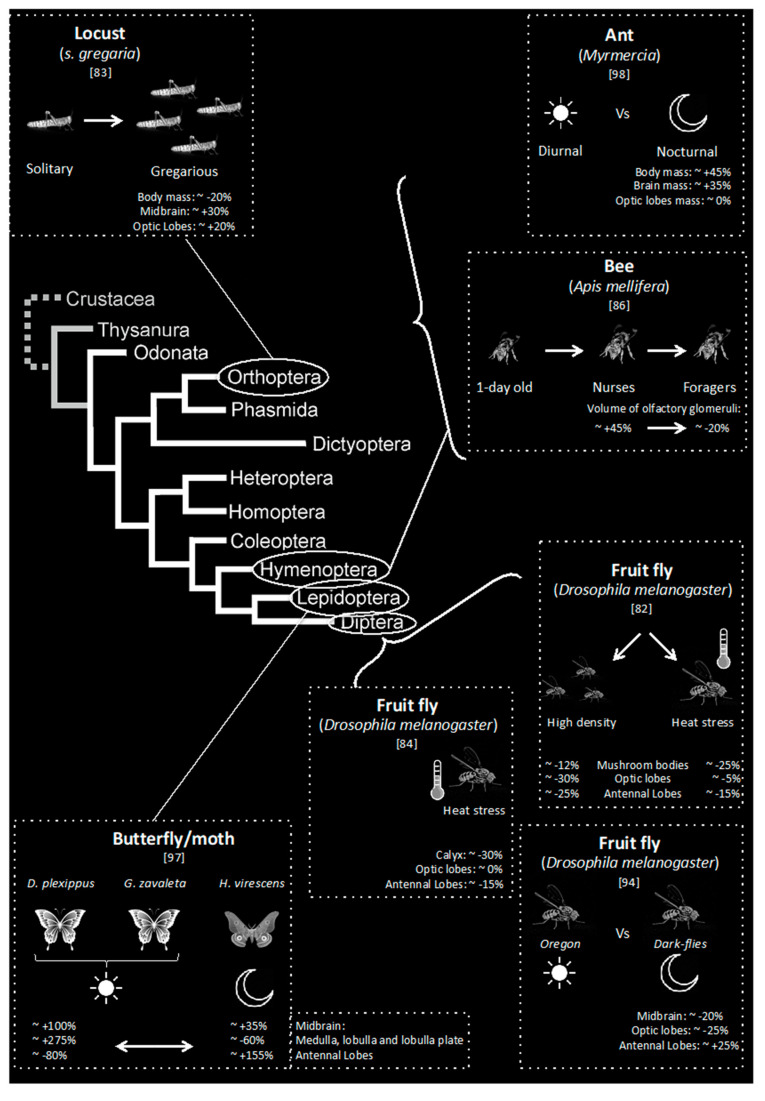
Examples of studies on influence of environment on brain size.

**Table 1 insects-12-00461-t001:** Summary of studies investigating effects of environment on brain size in insects.

Type	Species	Environmental Changes	Effects	Refs
Developmental modifications	Fruit flies	Social isolation, deprivation antennal input	Reduction in the number of fibres at the mushroom bodies	[79]
		Extreme larval crowding	Absolute volume of calyx, optic lobes, central brain and central complex increased	[80]
		Heat stress	Absolute volume of mushroom bodies reduced	[81]
		Extreme larval crowding	Absolute volumes of antennal lobes, optic lobes and central complex reduced	
	Desert locusts	Aggregation	Gregarious locusts have larger brains (larger midbrain)	[82]
	Fruit flies	Heat stress	Absolute volume of antennal lobes, calyx and pedunculus reduced	[83]
	Honeybees	Foraging experience	Bigger olfactory glomeruli in nurses than foragers	[84]
		Experience (foragers vs. nurses)	Absolute volume of mushroom bodies increased	[85]
	Ants	Dark and excavation (reduction of behaviours)	Reduction of medulla and total brain	[86]
	Fruit flies	Rearing in darkness	Absolute volume of optic lobes reduced	[87]
		Rearing in darkness	Absolute volume of mushroom bodies and central complex reduced	[88]
	Ants	Light exposure	Mushroom body calycal growth and reduction in microglumeruli numbers in the visual and olfactory input regions of the calyx	[89]
	Bumble bees	Presence of visual stimuli	Relative volume of antennal lobes and mushroom bodies reduced	[90]
	Fruit flies	Light enrichment	Whole brain volume increased. Absolute and relative volume of optic lobes increased. Absolute and relative volume of antennal lobes decreased	[91]
	Crickets	Complex environmental and congeneric stimulations	Increased number of newborn cells in their mushroom bodies	[92]
		Enriched sensory and social conditions	Enhanced neuroblast proliferation in the mushroom bodies	[93]
Evolutive adaptations	Butterflies, moth	Diurnal vs. nocturnal	Inversed investment in visual and olfactory systems: diurnal species invest more in vision whereas nocturnal species invest more in olfaction	[94]
	Ants			[95]
	Hawk moths			[96]
	Ants	Desert vs. forest	Number of glomeruli in the antennal lobes reduced	[97]
		Light vs. dark	Larger eye structure and visual neuropils in workers performing tasks in light	[98]

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
