# Peer review of "A Review of Effects of Environment on Brain Size in Insects"

_insects, 2021, doi:10.3390/insects12050461_

Round 1
Reviewer 1 Report
In “A vision of effects of environment on brain size in insects” the author summarizes relevant previous works on brain size in insects making a nice parallel with studies in vertebrates. This review focuses on untying brain size and intelligence and on describing some of the effects of environment on vision and olfaction brain sensory structures, and how this might influence overall brain size.
It is an exciting (and largely discussed) topic and the author presents an interesting hypothesis about how the relationship between olfaction and vision might determine brain size. However, I find the manuscript in its current form fails to present enough evidence supporting this hypothesis, missing to discuss some references that would point to the opposite conclusion. Additionally, there are some conceptual errors about isometry, proportionality, linear relationships and social behaviours as cognitive capabilities.
In general, despite the errors and missing references, I would say that while the digestion of the current knowledge presents a fair summary of the previous work, this review does not clearly provide an overview of open questions and possible new research avenues to answer them.
In the following, I will comment section by section on minor and major changes that I think would improve the manuscript, and I will detail changes that I think might help make it useful for researchers from this and other fields.
Simple summary:
Line 10: This might be a very minor comment, but “Researchers started to show that” is probably not necessary in this sentence. The use of this past tense requires a temporal reference that is missing (when?).
Line 11: maybe use “individual” instead of “personal” as the manuscript is talking about insects (as written in line 122).
Brain size:
Line 31: Does the author mean “allometric” instead of “isometric”? In Jerison, 1955, the relationship is described as allometric, and the log-log equation shows that there is an allometric relationship.
Line 32: I don’t think we can talk about proportionality when the relationship is allometric.
Lines 33-34: Why is it relevant to give these two equations that are so similar? I suggest to explain or to simplify.
Lines 34-36: While paradigmatic, this comment seems a bit disconnected to the previous description of the equations, which have been done using different species and not within the same species. Also, it disconnects the following sentence about disparities in the linear relationship with the equation itself. Maybe it would be better to remove it or to relocate it somewhere else.
Line 36 (and same comment for lines 110, 111): I suggest “([6] but see [7])” instead of “([6] but [7])” for better readability.
Lines 36-37: What would be a “perfect linear relationship”? Also, it is important to be careful here: there is not a linear relationship between brain and body mass, but between their logarithmic values.
Lines 38-39: I think it would be more clear to say something like “when analyzing the relationship between brain and body size by animal orders, the slope varies ranging from 0.24 to 0.81”.
Line 40: As said about lines 36-37, it needs to be clear that we are discussing about the log-log equation, and not about a linear relationship between brain and body mass.
Lines: 41-42: Social behaviours might require (or not) complex cognitive capabilities, but they are not an example of complex cognitive capabilities.
Line 43: Instead of “ecology”, which is probably too broad, I suggest “ecological drivers”.
Lines 48-49: Again, the social behaviour is not a cognitive capability.
Lines 49-51: the use of “introduce” in this aim seems to indicate that it is the first time that this idea is presented. Consider using other expressions such as “highlight”, “further develop” or others.
Haller’s rule, brain size in insects:
Line 55: Reference 25 is the same as 3 (Armstrong, E. Relative brain size and metabolism in mammals. Science (80-. ). 1983, 220, 1302–1304).
Line 63: This sentence suggests there would be a definition of intelligence that does not appear later. Also, maybe the real question is not how to measure intelligence in general, but if intelligence can be measured anatomically.
Line 64: In line with previous comment, consider mention that the measure is anatomical.
Line 68: I suggest “[…] whereas in mice it is […]” instead of “[…] whereas that in mice is”, and “(twice smaller)” instead of “or twice smaller” for better readability.
Effect of environment on brain size in vertebrates:
In this section, it would be useful to have a brief description on what the author means by environment effects and what kind will be discussed.
Line 93: Reference 46 is on insects (wasps) while this section is about vertebrates, did the author really mean to add it here?
Line 111: [for review 79], does the author mean ref. 80?
Lines 115-119: I suggest simplifying these sentences as “All these studies support Hebb's hypothesis*, which predicts that training, or differential experience, induces neurochemical changes in the cerebral cortex, and a differential cortex weight between those individuals that received this training, or experience, and those that did not [82].”
* Consider citing Hebbs, 1949.
Effect of environment on brain size in insects:
In this section, the different paragraphs and environmental effects seem a bit disconnected, which make this section to look as a simple enumeration of facts. As in the previous section, a brief introduction of the environmental effects discussed (social rearing context, experience, temperature, light conditions…) might help to better follow the point of the author. But other options are possible, such as smoother transitions between paragraphs.
Line 131: Consider adding a description of the central complex as done for the mushroom bodies and antennal and optic lobes.
Lines 126-128: In Heisenberg et al., 1995, other regions, as the optic lobes (at least medulla and lobula) also increased in the higher density group. Mentioning this might show better the contrast with Wang et al., 2018, as otherwise it could just be a different regional investment.
Lines 126-135: It could be interesting to discuss why this could be happening. Could it be an effect on food availability or competition? Also, at least in Heisenberg et al., 1995, socially deprived and socially reared are in different cage conditions, which might affect experience.
Line 145: I might be wrong, but it seems that Withers et al., 1993 found no difference in the Kenyon cell region between foragers and nurses.
Lines 146-149: It would be interesting to have a discussion about why the results by Withers et al., 1993 and Durst et al., 1994 might be different (as done for Heisenberg et al., 1995 and Wang et al., 2018). Could it be the age of the bees? the subregions that were analyzed?
Line 150: Mentioning the role of hormonal controls here requires more context. Also, reference 86 is on crickets; it seems a bit out of place in this sentence about bees.
Line 151: Consider relocating this sentence and giving more context, especially if there is an interest in citing the study on crickets.
Line 153: “D. melanogaster” instead of “Drosophila melanogaster” as it has already been mentioned in the manuscript.
Line 156: “D. melanogaster” instead of “Drosophila melanogaster” as it has already been mentioned in the manuscript.
Line 159-161: While interesting, I am not sure this description of the different choices is relevant here.
Line 163: “D. melanogaster” instead of “Drosophila melanogaster” as it has already been mentioned in the manuscript. Also, melanogaster is written melanogaster.
Lines 165-171: Given that the two studies measured different regions and in different species, this discussion is less relevant that one between the results by Barth and Heisenberg, 1997 and by Jones et al., 2013, as both measured mushroom bodies. In Sheehan et al., 2019 (Sheehan, Z. B., Kamhi, J. F., Seid, M. A., & Narendra, A. (2019). Differential investment in brain regions for a diurnal and nocturnal lifestyle in Australian Myrmecia ants. Journal of Comparative Neurology, 527(7), 1261-1277), nocturnal ants also have mushroom bodies that are bigger (relative to brain mass) than diurnal ants.
Line 170: If this discussion remains, then “that effects occur” or “that effect occurs”.
Lines 156-178: Consider reorganizing content to discuss in a first paragraph Barth and Heisenberg, 1997 and the similarity with Özer and Carle, 2020. Then the disparity with Jones et al., 2013. And later, maybe in another paragraph, the similarities between Jones et al., 2013 and Özer and Carle, 2020 to introduce the idea of the trade-off between vision and olfaction.
Balance between vision and olfaction:
This is where I find the author should make a bigger effort explaining why this balance would be important for brain size and if that opens new research avenues and interpretations in the field. Also the evidence for this balance should be more strongly discussed and supported.
Lines 211-212: In another work cited in this manuscript (Muscedere et al., 2014), in ants, there was found no evidence of this trade-off, as also happened in bats or insectivores in Barton et al., 1995. This contradictory results should be discussed, especially if the author would like to make a point about the visual system affecting brain size.
Line 213-214: I don’t think the reader is ready to understand this association between visual inputs and brain size at this moment. This idea would be better placed after the evidence provided later.
Line 219: I am not sure if the author relates size of visual system to total brain size or just to the size of certain regions. But, while I am not an expert in fruit fly brain litterature, it seems that references 92 and 93 do not correlate total brain size to the investment in visual systems. The references 95 and 103 do show a correlation between brain size and optic lobes, and there is similar correlation in Arganda et al., 2020 (Arganda, S., Hoadley, A. P., Razdan, E. S., Muratore, I. B., & Traniello, J. F. (2020). The neuroplasticity of division of labor: worker polymorphism, compound eye structure and brain organization in the leafcutter ant Atta cephalotes. Journal of Comparative Physiology A, 206, 651-662). Also, it should be discussed that in Sheehan et al., 2019, the authors found a negative correlation between optic lobes and antennal lobes comparing diurnal and nocturnal ants, but the relative brain size did not differ between them.
Lines: 220-221: I think I do not understand what the author meant by “This conclusion was taken up by Farris et al. [105] by commenting on Barth et al.’s experiments”. Farris et al., 2016 does not comment on Barth et al.’s experiments. It discusses the role of novel visual requirements for behavior and the effect on mushroom bodies (which receive visual inputs). Also, how Muscedere et al., 2014 helps to build this conclusion is not clear to me, unless it is based on the result that mushroom body size can vary independently of brain size. This sentence should be rewritten for clarity.
Lines 222-223: Justify further this hypothesis.
Line 223: “Stieb” instead of “Stied”.
Line 232: Could the author find evidence in other works with similar species for this hypothesis?
Conclusion:
Lines 235-237: Before discussing cognitive abilities, those animals are also very different in body size, which the manuscript starts recognizing as a fundamental predictor of brain size. Is the author talking about relative brain size or absolute brain size?
Line 246: absolute or relative brain size? What about the EQ introduced before?
Lines 248-249: Better develop the implications of neuron size and evolution.
Figure 1:
This figure is mentioned twice in the text: in lines 74 and 237. In line 74, it is used when discussing Haller’s rule. It could be then useful to have also a measure of the average size of each animal, so the reader can have an idea (if possible) of that rule.
It could also be considered to add a schematic outline of the insect brain to show some of the different functional regions that are discussed in the manuscript (mushroom bodies, optic and antennal lobes…).
Table 1:
As presented in the manuscript, the table is divided in two pages and it is difficult to read. It would be better to find a way to have it in a single page.
In the caption for the table, it would be useful to have an explanation of what it is meant by grouping them in “Plasticity” and in “Evolution”.
I find useful to see references as “Technau (1984)” but having also the reference number (as in the rest of the manuscript) might facilitate to find the reference in the text and in the corresponding section. Additionally, the reference “Julian & Gronenberg (2002)” is not mentioned in the manuscript and it is missing in the reference section.
In the part “Evolution” related to ants and light conditions, some references can be also interesting to discuss, such as Sheehan et al., 2019 and Arganda et al., 2020.
References:
Names of species are not in italics in many references.
Author Response
Dear Reviewer,
I really thank you for your ideas and suggestions.
Please find the modifications that I brought to the manuscript from your comments and the comments from other reviewers in the file attached.

Reviewer 2 Report
Thank you for the opportunity to review “A vision of effects of environment on brain size in insects” for Insects. While not introducing an especially novel take on the evolution of large brains, it does provide a summary of recent contributions to the field. It also provides a new prospective that has not been clearly articulated previously: that there is likely interplay between the requirements of different sensory modalities during the course of evolution. I believe a review such as this one is timely and important.
That said, however, I believe a major revision of the present manuscript is required. Many sentences are very hard to understand, and the content seems to be somewhat scattered. I would expect the language to be improved before I could recommend publication, and I would also recommend an attempt to clean up the organization a bit. Parts of it seem almost encyclopedic with lists of details and references. Finally, I have a few questions and suggestions regarding the content. Details follow.
- Line 2: The title is very confusing. Was it intended to be, “A review of effects of environment on brain size in insects”?
- Line 7: This first sentence is also confusing. What was considered as a measure of intelligence, brain size?
- Line 9: Insert comma after “such behaviours”.
- Lines 34-35: Double (or triple) negative, and confusing. Did you mean “…none can disagree…”?
- Line 44: You might also site an insect example, since this is an insect paper: e.g., Farris & Roberts (2005).
- Lines 95-102: I think a brief discussion of the loss of the other sense might be interesting. You discuss the loss of vision in animals that evolved in caves. What happens in the brains of secondarily anosmic insects such as cicadas whose mushroom bodies and/or antennal lobes are reduced or absent? Also, how do you think the addition of another important sensory modality such as audition plays into the interplay between olfaction and vision (e.g., crickets, cicadas)?
- Line 115: Change “experienced” to “experience”.
- Lines 141-142: Move “honeybees (Apis mellifera) to this first sentence where they are first mentioned.
- Lines 141-151: Withers et al. (1993) showed clearly that there were differences in the MB calyces between honeybee nurses and foragers. Durst et al. (1994) showed that foraging experience contributes to the growth of MB calyces in age-controlled bees. However, Fahrbach et al. (1998) showed that individually caged bees exhibit growth of the MB calyces with age. (This is also suggested in Withers et al. (1993) but not as clearly.) It is true that the life-long calycal growth in honeybees is mostly attributable to foraging experience, but age plays a role too. It might also be important to point out that the mushroom bodies of adult honeybees are postmitotoc; all exhibited growth is due to increases in the dendritic complexity of the Kenyon cells as opposed to any neurogenesis (Farris et al. 2001, which again shows both age- and experience-based growth).
- Lines 156-161: The opposite is true too. In desert ants (Steib et al. 2010), light exposure triggers MB calycal growth.
- Line 163: Correct the spelling of melanogaster.
- Line 169: Change “bodies” to “body”.
- Line 180: Please add reference numbers to the References column in Table 1 so that they can be found in the References section of paper.
- Line 180: Do the vertical bars in the middle of Table 1 signify something?
- Line 186: Is “insectivore” referring to bats or primates?
- Line 189: Similarly, nocturnal, frugivorous bats or primates?
- Lines 226-229: Please elaborate. How is relying on smell a better strategy when dealing with obstacles such as trees or leaves?
- Lines 247-248: Incomplete sentence. Did you mean to use the conjunction “whereas”? If so, it needs a preceding comma, and the dependent clause needs a verb. I do not understand the sentence as is.
Very minor comments/suggestions you might consider:
- Line 8: Would “associated with” be better (and consistent with line 17)?
- Line 29: “Annihilate” is a rather strong word, no? Maybe “nullify” instead?
- Lines 65-66: Consider rewording: “…the ratio between brain mass observed and that predicted from body size…”.
- Lines 113,114: Isn’t the plural of “fish” (as in a species) “fishes” (many species of fish) and the plural of “fish” (as in the individual animal) just “fish” (many individual fish)? Or are these just regional differences (like “connection” vs “connexion”)?
- Line 131: It’s likely the audience of Insects knows the term “central complex” without defining it. But you give very brief descriptions of mushroom bodies, optic lobes, and antennal lobes above.
- Line 174: Add “the” before “olfactory system”.
- Line 234: Add “the” before “visual system”.
- Line 248: Add “the” before “neuron”.
- Line 254: Change “fro” to “from”.
- Line 255: Change “who” to “whom”.
- Line 452: BB&E shows your paper as 2019, not 2020. (I did not look closely at the details of many of the references, but I recommend you double-check them all.)
Author Response

(The authors gave the same response as above.)

Reviewer 3 Report
An interesting overview of the important issue of the evolution of insect brain size. The author has chosen an original direction for studying this issue and made a short but rather complete review of modern works on this topic. The only weakness of the manuscript is the graphical presentation of the material. The only drawing is uninformative. In such a review, I would like to see summary graphs based on numerous literature data, quantitative characteristics superimposed on phylogenetic trees, etc.
Author Response

(The authors gave the same response as above.)

Round 2
Reviewer 1 Report
I recognize the substantial effort made by the author to reply to my remarks (and those of the other reviewers), despite that, I still have some concerns and, in my opinion, the new version would benefit from more refining work to be an enjoyable and useful revision on how the environment affects brain size in insects. While some parts have improved, I find that some others have become repetitive and a bit messy. Additionally, I might have missed them, but I don’t find some of the changes that the author points to have done (I will mention them in the corresponding lines or sections). I hope my next comments can help the author to improve the manuscript.
Simple summary:
I find the current version is satisfactory
Abstract:
I find the current version is satisfactory
Brain size:
I think this section has improved a lot and just needs some minor changes:
Line 31: I would remove “for example” as “such as” already implies “an example”
Lines 34-35: I understand now that the author would like to make the point that new studies (I imagine that they used new species data) obtained a very similar equation. If that is the point of the author, it would be better to highlight it. Maybe as: “or more recently (50 years later and using substantially more species)”.
Line 38: I would write “this relationship” instead of “the relationship”, to make a clearer reference to the log/log analysis but without being repetitive
Line 40: I am not sure what the author means by “animal’s relative cognitive abilities”
Line 42: I suggest to write “as posits by the social brain hypothesis” instead of “: the social brain hypothesis”
Line 43: “are better predictors” instead of “is a better predictor”
Lines 44-47: Part of this sentence is a bit repetitive with the previous one. Maybe it would be less repetitive phrased as: “More generally, a part from ecological factors, other hypotheses considering energetic, life history and behavioral factors has been highlighted to explain these disparities and to support anticipated […]”
Line 49: “involved” instead of “implies”?
Haller’s rule, brain size in insects:
I also think this section just needs minor changes:
Line 56: […] “that the increase in brain weight is inferior to the increase in body weight […]”
Lines 62-63: I think this sentence would be better placed starting the following paragraph
Line 68: As mentioned in my previous revision, I would add parenthesis to “(or twice smaller)” for better readability.
Effect of the environment on brain size in vertebrates:
I also think this section just needs minor changes:
Line 91: “have driven” or “might have driven”? At least for the Social Brain Hypothesis (if that is what the author implies with “increase in cognitive demands”), evidence is still under debate. Also, even if the information is on the title of the section, I suggest adding “in vertebrates” after “evolution of brain size” to focus the topic.
Line 94: I would add “or” before“scarcity of food”.
Line 112: While I don’t think it is important, the author said “[ref] but [ref]” had been changed by “[ref] but see [ref]”, however it is not the case.
Lines 115 and 117: I would remove “the presence of” before “a maze to find food” to make simpler sentences.
Effect of environment on brain size in insects:
This section is one of those that need a bit more work. I think the ideas are there, but I think the author might improve this section by building simpler sentences and avoiding unneeded repetitions.
My suggestions are:
Lines 125-126: (minor comment) Definition of the MB inside parenthesis. Consider qualifying the MB as an “integration area” or “higher order processing area”.
Line 126: (minor comment) “reduced” instead of “reduces” as it is describing a result in literature.
Line 128: (minor comment) “had” instead of “have” as it is describing a result in literature.
Lines 128-131: Consider simplifying this sentence as “[…] that larval density (affecting social experience) had a positive effect on the number of Kenyon cell fibers (intrinsic neurons of the mushroom bodies), the absolute volume of mushroom body calyces and of optic lobe medulla and lobula (brain areas processing visual information)” if it is correct.
Lines 138-139: I think this sentence could be removed as it is not providing new information. If the author would like to discuss the cricket study, it could be probably better placed in the next paragraph.
Lines 142-147: “larger sizes in all subcompartment”, while true, is a bit confusing regarding the results of the study. The brains are larger in gregarious locusts, but the relative volumes of each subregion change in the two phases. I suggest to remove the sentence “larger sizes in all subcompartment” and to describe better later which higher integration regions are relatively bigger in the gregarious forms.
Line 148: “had” instead of “has”.
Line 150: “were” instead of “are”.
Line 152: Why would it be obvious?
Line 154: I suggest “honeybees (Apis mellifera)” instead of “bees” and only use honeybees in the successive mentions.
Lines 158-161: I suggest to simplify these sentences and merge them into one “In particular, Withers et al. [89] analyzed the anatomical changes associated to age and the naturally occurring behavioural transition from nursing to foraging tasks, and showed that the size of brain structures depended on foraging experience in honeybees (Table 1). “
Line 172: “and not due to neurogenesis”, I am not sure that refs 89, 90 or 91 discuss neurogenesis.
Line 192: I am confused about the use of the word “experiment”, because it seems it is suggesting that the study of Sheehan et al (2019) observed changes due to experimental manipulations, while it simply analyzed the anatomical differences between nocturnal and diurnal ants. Therefore, I agree this study’s results go in the same direction as Özer and Carle (2020) but one discusses evolutionary differences between species and the other, experience related plasticity within the same species.
Lines 192-194: Why not to discuss too studies on nocturnal and diurnal moths that go in the same direction?
Line 196: I suggest adding “in bumblebees” after “with Jones et al.’s [97] study, to better remind the reader about the study.
Lines 196-198: Sheehan et al (2019) did not study visual enrichment in species that lived in the dark.
Balance between vision and olfaction:
In my opinion, this section would benefit from trying to avoid being too repetitive when explaining again previously cited studies (for example, Ozer and Carle, 2020).
Additionally, in my previous revision, I mentioned that a work cited in the manuscript showed that there was no evidence of the trade-off between olfaction and vision in ants (Muscedere et al., 2014). In the current response to the reviewers, the author says to have discussed this in the new version of the manuscript. But while it is mentioned in this section the fact that in insectivore mammal species there is also not such an evidence, the article on ants is also not discussed and it has disappeared from the manuscript. I think these results that contradict the main hypothesis of this manuscript should be better discussed (here or in the next section).
Also the author might want to use the sample of dragonflies and mayflies, that have large optic lobes and underdeveloped or absent antennal lobes.
Line 218: it would be useful for the reader to know the lifestyle of the glasswing butterfly to understand its particular brain investment in comparison with diurnal and nocturnal butterflies and moths.
Lines 235-238: This seems to have been already explained in previous sections. Considered simplifying this explanation and also recognize that the results by Keesey et al (2019) and Ozer and Carle (2020) operate at different levels (evolution vs species plasticity).
Effect of environment on brain size through this trade-off?:
It would be probably more helpful to keep “vision and olfaction trade-off” in the title.
Lines 265-266: I think at this point the reader might not really understand which brains are not measured, as some Cataglyphis species have their brain already measured.
Lines 272-273: Does the author mean that the volume of OL and AL are similar within an ant brain, or that among ant species, the sizes of those structures are similar. To my knowledge, both affirmations would be wrong.
Lines 282-288: Discussing dragonflies and mayflies might help here, as their brains have been imaged.
Lines 289-299: Discuss also here the work in ants by Muscedere et al (2014)
Conclusions:
I think this section reads better now and provides more suggestions to the reader.
Line 305: “others” instead of “other factors” (suggestion)
Lines 306-307: consider using “might be” instead of “are”, as it is not proven that more complex social interactions are behind the evolution of brain size.
Line 325: I am not sure what is implied by the author when using the word “everything”. Its use here might be seem as a bit empty. Maybe more context or focus are required.
Figure 1:
Maybe better write “Workers’ length” instead of “Workers length" when appropriate.
Table 1:
It looks better now, but it is still divided between two pages, making it more difficult to read.
Author Response
I would like to thank you for the positive comments towards my manuscript. The changes that were made are included in the file.

Reviewer 2 Report
In the revision of “A review of effects on environment on brain size in insects”, I believe the author has addressed the majority of my concerns adequately. My one remaining concern is that the language is still very messy. I did not have time to hunt down every instance for correction, but I have listed some examples below. I would expect the manuscript be revised one last time after very thoroughly being proofread. I find the grammatical errors districting, reducing my focus on the point of the paper. Some specific comments:
- Line 43: subject-verb agreement: “…ecological drivers, including diets, are better predictors…”
- Line 44: subject-verb agreement: “…other hypotheses have been highlighted…”
- Line 128: subject-verb agreement: “social experience has a positive effect…”
- Line 135: missing article: “…that play a crucial role…”
- Line 144: Consider replacing the ellipses (which fills in for missing words) with “etc.”
- Line 148: Replace the first comma with “and”.
- Line 148: Remove the last comma: “…a single daily 39.5°C pulse for 35 min has a…”
- Line 150: missing article: “…particularly of the calyx and pedunculus…”
- Lines 150-151: Do you mean, “… but the optic lobes and central complex are not effected…”?
- Line 207: Change “mammals species” to “mammalian species”.
- Lines 257-258: Move the definition of olfactory glomeruli to the first use of the term on Line 162.
- Lines 279-280: Change to “… which may explain the lack of difference…”
- Line 284: run-on: Either replace the comma with a semicolon between clauses or make two sentences.
- Lines 301-302: subject-verb agreement: “…environment…plays a role…”
- Line 322: Change “neurons size” to “neuronal size”.
- Line 322: missing article: “…a larger brain…”
- Please rephrase the last sentence. Maybe “…mini-brains exhibit unsuspected complex cognitive capabilities…”?
Other thoughts:
- Figure 1: What are the sources of these brain and insect images? Are these the author’s own images, or do these come from other papers?
- Line 315: Maybe rephrase so the two sentences in the Conclusions section do not both start with “In this review”.
Author Response

(The authors gave the same response as above.)

Reviewer 3 Report
I had only a couple of general comments, one about the uninformativeness of the picture, the second about the need to summarize quantitative data. And I am very surprised at the answer. Nothing has changed in the manuscript. The author only admits that the analysis of the literature data needs to be carried out. It seems to me that this analysis needs to be done in this review. Without it, the main conclusions are too speculative.
Author Response

(The authors gave the same response as above.)

Round 3
Reviewer 1 Report
In this new version of the manuscript, I can appreciate changes that have made the text clearer. However, it seems to have some small contradictions between the main hypotheses and what it is actually presented in the article. For example, the abstract final remark is about how the brain size might be affected by the relationship between olfaction and vision, but now the manuscript is not presenting this hypothesis so strongly and provides other alternatives (which I think is great!) Also, the main point of this revision is about total brain size, but often, subregional changes of the brain are discussed. While it makes sense, as the idea is that environment might affect brain structures, this has to be carefully introduced. I think this comment is related to the apparent complexity of Figure 1’s new tables. Figure 1 shows more information now, but in its current form, I find it is not clear how that information might guide the reader to the conclusions of this article.
I have some suggestions that might help to focus a bit more the content of the manuscript:
Simple summary:
Line 13-14: As the author has not done any analysis of the data, which I think it is one of the main criticism of the Reviewer 3, I suggest to write “I also discuss evidence about how the environment has an impact on sensory systems and influences brain size”.
Abstract:
Line 23: Because the author now discusses further how other sensory modalities might also play a role and that a trade-off between olfaction and vision might not be in place, I suggest to rephrase the sentence as “that the role of environment might be mediated through the relationship between olfaction and vision”.
Brain size:
I think this sections reads well now.
Haller’s rule, brain size in insects:
Line 53: (minor comment) I suggest to remove “also”.
Line 64: (minor comment) I suggest to use “:” instead of “,” in “[…] (EQ), the ratio […]”.
Line 73-74: (minor comment) I suggest removing “which means that smaller individuals possess relatively bigger brains” as it was already explained before.
Line 78-79: (minor comment) I suggest saying “(with the brain occupying 8.2% of the body in this case)”.
Effect of environment on brain size in vertebrates:
Line 91: (minor comment) I suggest removing “are factors that” to avoid repetition of the word “factor” and to simplify the sentence.
Line 125: (minor comment) as mushroom bodies is in plural should the information inside the parenthesis be also in plural? (“higher order processing areas in the central brain that integrate”)
Line 128: (minor comment) Can we be sure that larval density is affecting the number of Kenyon cell fibers through social experience? It seems to me that it is more a hypothesis than a conclusion. Maybe it could be said “larval density (which might affect social experience)”.
Line 161: (minor comment) In general, I don’t think it is necessary to mention again the name of the species after the common name if it is referring to the same species already mentionned, but if it is preferred like that, at least I would suggest writing it in the abbreviated form “A. mellifera”.
Lines 173-176: (minor comment) This paragraph is probably better placed as part of the previous one.
Balance between vision and olfaction:
Line 232: I believe Godyris zavaleta is diurnal. I find that adding this information will help the reader to understand better the following discussion.
Line 236: (minor comment) I suggest adding a comma in “ […] with behavioural observations, suggesting […].
Line 237: (minor comment) For the clarity, I suggest rephrasing as “[…] may be more important in guiding a suite of behaviours in the glasswing than in the monarch”.
Effect of environment on brain size through this vision and olfaction trade-off? :
Lines 262 and 264: (minor comment) “easily” used twice too close in the text.
Line 278: I am not sure I understand what the author means by “the species used in these studies does not perfectly fit”.
Line 279: I suggest “avenue” instead of “way”, which in my opinion leads to think more in “how to do research” than in “what to look for”.
Vision and olfaction trade-off a universal rule? :
Line 285-286: I suggest “the ratio between optic and antennal lobe volumes in ants is very low”.
While I understand the answer of the author regarding this topic, I still think it can be interesting to discuss, or at least to mention, Muscedere et al (2014) in this section, as they also contest the existence of a trade-off, at least in ants (“There was no evidence of a direct trade-off between investment in peripheral visual and olfactory processing neuropils, which could have reduced variation in C/AO among species with different visual abilities: the relative sizes of the ALs and OLs (with respect to all other brain regions) were not significantly correlated across species”) and they find that larger ants tend to have larger brains and larger investment in vision (“(iii) larger bodied (and brained) ants therefore tend to have larger eyes and OLs; thus (iv) larger brained species tend to have lower C/AO ratios, driven by the effect of their relatively larger OL volumes”) – as it was mentioned in previous versions of this manuscript. However, I think this is the decision of the author, especially considering how this could be explained and discussed.
Conclusion:
Line 321-323: This sentence is a bit messy. I suggest rephrasing it.
Line 329-330: I am not so sure the main discussion of this manuscript is about studies contradicting the relationship between brain size and intelligence, but about other factors affecting brain size.
Figure 1:
I find it much messier now and I think it needs some guidance (the figure caption could be used for this purpose). It is not clear to me what information should be extracted from the tables, and the fact that there are different headers in each table adds more confusion. Maybe the author should consider plotting some figures with these data, or present them in another complementary table or subfigure. I find that, in the current format, the figure is too confusing and charged.
Table 1:
The title reads “Summary of studies investing effects of environment on brain size in insects”, was it meant to be “investigating” instead of “investing”?
Additionally, there are some cut words after “Mushroom bodies*1 calycal growth and reduction in microglumeruli* numbers in the…”. Also: *1 Mushroom body (I believe it has to be in singular here), *2 microglomeruli
Author Response
I thank you for these new comments and for your precious time. As previously, I included the changes in the file attached.

Reviewer 2 Report
I think the language of the manuscript is largely improved. I do have a handful more comments regarding wording, however (see below).
One thing I'm still confused by is the vertical bars in the last few rows of Table 1. I think they should be explained in the caption or in a footnote to the table. Also note that the 11th row under Effects is mostly missing, at least in my PDF copy.
My biggest concern now is Figure 1. It's mentioned only on line 73. I guess it should be illustrating how insects follow Haller's rule. Figure 1 doesn't really do this, however. There are just random brain sizes and body lengths spread out through the figure. It's not easy to compare them. It would much better if you graphed these values against each other in some way. I'm also not really sure how the new data tables on brain regions contribute. Those tables aren't mentioned anywhere. Finally, the figure needs to be cleaned up. The tables are randomly distributed, never lining up, sometimes butting up against one edge of the figure or the other.
One other minor correction: On line 113, is the review not reference 76 instead of 77?
Minor language concerns:
Line 41: Do you mean "as posited by"?
Line 44: Replace "a part" with "apart".
Line 45: Replace "has" with "have".
Line 69: Add "the" before "human brain".
Line 96: Replace "determinator" with "determinant".
Line 103: I think "in association with" is better.
Line 113: Replace "or" with "and".
Line 116: I think the plural of fish (individual animals, as opposed to multiple species) is "fish".
Line 159: "association with".
Line 178-181: This sentence is long and confusing. I think you should split it into two sentences. End the first sentence immediately after the [90] reference. Then at the beginning of the next sentence, replace "which" with "These decreases". (If this is in fact what you're trying to say.)
Line 185: Replace "seems" with "seem".
Line 191: Maybe: "This seems in line with Montgomery..."
Line 199: Replace "have" with "has".
Lines 210-212: This sentence has always confused me. Maybe consider this wording: "... Barton [99] showed three years later that in primates, the bigger the visual system, the bigger the brain and, therefore..."
Line 273: Replace "partner" with "partners" (or replace "sexual partner" with "mates").
Line 285: More confusing language. How about: "... the fact that the ratio of the volume of the optic lobes to the volume of the antennal lobes is low in ants."
Line 288: Replace "to the detriment to" to "to the detriment of".
Line 291: Awkward. Maybe: "... if this hypothesis is not verified in ants, it needs to be..."
Line 310: Change "enables to determine" to "enables the determination of".
Line 336: associated with.
One last thought: As I was about to send this report, I received notification of a new paper: "Reversible plasticity in brain size, behaviour and physiology characterizes caste transitions in a socially flexible ant (Harpegnathos saltator)" (Penick et al., 2021, Proc R Soc B 288: 20210141). I haven't had a chance to look at it, but it might be relevant to your review.
Author Response

(The authors gave the same response as above.)

Reviewer 3 Report
A rigorous analysis of the data never appeared in the manuscript, but the article can be accepted for publication in this form by the decision of the editor.
Author Response

(The authors gave the same response as above.)
